# Long-Term Effects of Subthalamic Stimulation on Motor Symptoms and Quality of Life in Patients with Parkinson’s Disease

**DOI:** 10.3390/healthcare11060920

**Published:** 2023-03-22

**Authors:** Jiin-Ling Jiang, Shin-Yuan Chen, Sheng-Tzung Tsai, Yu-Chin Ma, Jen-Hung Wang

**Affiliations:** 1Department of Nursing, Tzu Chi University, Hualien 97004, Taiwan; myc600221@gms.tcu.edu.tw; 2School of Medicine, Tzu Chi University, Hualien 97004, Taiwan; paramananda@gms.tcu.edu.tw (S.-Y.C.); flydream.tsai@gmail.com (S.-T.T.); 3Department of Neurosurgery, Hualien Tzu Chi Hospital, Buddhist Tzu Chi Medical Foundation, Hualien 97004, Taiwan; 4Department of Medical Research, Hualien Tzu Chi Hospital, Buddhist Tzu Chi Medical Foundation, Hualien 97004, Taiwan; paulwang@tzuchi.com.tw

**Keywords:** long-term, motor symptoms, Parkinson’s disease, quality of life, subthalamic stimulation

## Abstract

Parkinson’s disease (PD) is a progressive neurodegenerative disorder affecting both motor functions and quality of life (QoL). This study compared motor symptoms and QoL in patients with PD before and at 1 and 5 years after subthalamic nucleus deep brain stimulation (STN-DBS) surgery in Taiwan. This study included 53 patients with PD undergoing STN-DBS. The motor symptoms improved by 39.71 ± 26.52% and 18.83 ± 37.15% in the Unified Parkinson’s Disease Rating Scale (UPDRS) part II and by 36.83 ± 22.51% and 22.75 ± 36.32% in the UPDRS part III at 1 and 5 years after STN-DBS in the off-medication/on-stimulation state, respectively. The Hoehn and Yahr stage significantly improved at the 1-year follow-up but declined progressively and returned to the baseline stage 5 years post-surgery. The Schwab and England Activities of Daily Living improved and sustained for 5 years following STN-DBS. Levodopa equivalent daily dose decreased by 35.32 ± 35.87% and 15.26 ± 65.76% at 1 and 5 years post-surgery, respectively. The QoL revealed significant improvement at 1 year post-surgery; however, patients regressed to near baseline levels 5 years post-surgery. The long-term effects of STN-DBS on motor symptoms were maintained over 5 years after STN-DBS surgery. At the same time, STN-DBS had no long-lasting effect on QoL. The study findings will enable clinicians to become more aware of visible and invisible manifestations of PD.

## 1. Introduction

Parkinson’s disease (PD) is the second-most common progressive neurodegenerative disorder and involves significant disability, discrimination, and stigmatization with a compromised quality of life (QoL). Levodopa therapy, a common pharmacological treatment for PD, is associated with the development of complications and presents major challenges for long-term use. Deep brain stimulation (DBS) is effective in controlling levodopa-responsive movement abnormalities in PD. The possible mechanisms include interference with neural signals, the desynchronization of abnormal oscillations, the alteration of inhibition and excitation within neural networks, and the modulation of neurotransmitter and hormonal signaling [1]. DBS offers several advantages over other surgical approaches for neuromodulation. These advantages include the capacity to titrate stimulation parameters to maximize benefits and reduce adverse effects and the opportunity to directly interface with the circuit pathology that drives overt symptoms [2]. A systematic review showed DBS was superior to the best medical therapy at improving impairment/disability, QoL, and reducing medication doses, but these benefits need to be weighed against the higher risk of serious adverse events [3]. Currently, subthalamic nucleus deep brain stimulation (STN-DBS) is the most preferred surgical procedure for PD [4,5]. Several randomized controlled trials of DBS, which utilized the Unified Parkinson’s Disease Rating Scale (UPDRS) part III, have confirmed its short-term efficacy on the motor symptoms of PD [6,7,8,9,10,11,12,13]. Numerous reviews and meta-analyses of the short-term (3 months) and long-term (36 months) efficacy of DBS in patients with PD have also been published [3,14,15,16,17]. In favor of DBS efficacy, significant improvements in the UPDRS score, 39-item PD Questionnaire (PDQ-39) QoL score, and levodopa equivalent daily dose (LEDD) were reported in a systematic review and meta-analysis of randomized controlled trials [3]. Another meta-analysis indicated a greater reduction in medication and improvement in off-phase symptoms following STN-DBS compared with globus pallidus pars interna-DBS [14]. Mahlknecht et al. demonstrated that there are signals from controlled long-term observational studies, suggesting that STN-DBS may delay some of the late-stage disabilities, including psychosis, falls, and institutionalization, and slightly prolong survival compared with matched medically managed patients [18]. However, the motor improvement does not necessarily mirror the improvement in QoL after DBS; post-surgery, some patients reported dissatisfaction despite improvement in motor functions [19].

Self-assessed health status may provide a more comparable predictor of outcome than many objective measures of health. The motor benefit, with a slight decline, reportedly remains significant up to 5 years after STN-DBS surgery [20]. Nonetheless, independent life and QoL, evaluated with the PDQ-39, showed no significant differences between STN-DBS and best medical therapy groups at the 12-year follow-up [21]. Additionally, four studies that used the PDQ-39 for QoL assessments consistently reported that the benefits observed 1 year after STN-DBS were lost by 5 years [22,23,24,25]. Therefore, the evaluation of QoL after STN-DBS surgery is of paramount importance. Careful follow-up observation is recommended to examine the neurological deficit and to modify the regimen of pharmacological treatment.

Most investigations regarding the effects of STN-DBS surgery on motor symptoms and QoL in PD patients were carried out in clinical settings in Western countries. Few have reported long-term outcomes following STN-DBS in Asian populations. The efficacy of STN-DBS surgery in Asian patients with PD requires investigation. The current study aimed to compare motor symptoms and QoL in patients with PD before and 1 and 5 years after bilateral STN-DBS surgery in Taiwan. The patients were assessed at the three time points using the UPDRS, Hoehn and Yahr (H & Y) stage, Schwab and England Activities of Daily Living (SEADL) scale, LEDD, and the PDQ-39 questionnaire.

## 2. Materials and Methods

The current study utilized a prospective, observational cohort design to evaluate changes in motor symptoms and QoL in patients with PD before surgery and 1 and 5 years after STN-DBS surgery.

### 2.1. Participants

A total of 79 consecutive PD patients who underwent STN-DBS surgery at the Tzu Chi medical center in Taiwan between May 2011 and September 2020 were enrolled in this study. Thorough neurological, neuropsychological, radiological, and systematic medical evaluations were comprehensively performed by a team. The diagnosis of PD followed the United Kingdom PD Society Brain Bank criteria [26]. Consequently, patients with little or no response from pharmacological therapy were considered poor surgical candidates. Other relative contraindications for surgery included significant psychiatric disturbances and dementia. The institutional review board approved the study protocol (IRB 097-32; Tzu Chi General Hospital, Hualien, Taiwan). All patients participating in this study signed written informed consent for STN-DBS surgery and the study’s evaluation procedure.

### 2.2. Surgical Procedure

Magnetic resonance imaging (MRI) brain was obtained before the STN-DBS surgery to assess for hematoma and edema and to verify lesion placement. Initial targeting was guided by the integration of MRI, with frames for stereotactic surgery, via specially designed computer software. DBS procedures were performed under local anesthesia to maintain the patient with clinical criteria intraoperatively. After placing the stereotactic frame, computerized tomography (CT) examination was performed to locate the coordinates of the anterior and posterior commissures. Computer programs allowed for subsequent simulation with the patient’s MRI of the precise trajectory and distance to target from the burr hole. An electrophysiological assessment of the activity of the targets was used to ensure proper targeting and placement of the electrode. The electrode was placed at the intended target with X-ray confirmation followed by intraoperative stimulation and characterization of stimulation effects. Brain CT was performed immediately after the surgery to confirm the implanted site and to explain procedure-related complications. After one week, the electrode cables were connected to the implantable pulse generator. Detailed surgical procedures are described in our previous studies [27,28,29,30].

### 2.3. Measures

#### 2.3.1. UPDRS

The UPDRS includes four subscales. Part I covers mentation, behavior, and mood. Part II rates activities of daily living. Part III is a clinician rating of the motor manifestations of PD. Part IV covers complications of pharmacological therapy. Each item is scored on a scale from 0 to 4 (UPDRS total range of scores, 0 to 199), and a higher score indicates a worsening in function [31]. The outcome measurements were changes compared to the baseline in the UPDRS part II, part III motor scores ranging from 0–108, and subscales for tremor at rest (items 20–21), rigidity (item 22), bradykinesia (items 23–26 and 31), posture and gait (items 28–29), and axial symptoms (items 18 and 27–30) in the off-medication and on-medication state with the stimulator on and UPDRS part IV (motor complications of therapy) at 1 and 5 years after surgery. The off-medication state was evaluated after the withdrawal of dopaminergic medication for at least 12 h, as defined by the core assessment program for surgical interventional therapy in PD [32]. Martinez-Martin et al. showed that the internal consistency of UPDRS was high (Cronbach’s alpha = 0.96). The inter-rater reliability was satisfactory (all the items yielded k > 0.40). There was a high correlation of the UPDRS with the H & Y staging (r = 0.71; *p* < 0.001) and some timed tests (finger tapping; arising from a chair). The convergent validity with the other PD rating scale (UPDRS) was very high [31,33].

#### 2.3.2. H & Y Stage

The H & Y classifies PD patients into five stages according to the body distribution of symptoms and dependency [34]. The scale is based on the two-fold concept that the severity of overall Parkinsonian dysfunction relates to bilateral motor involvement and compromised balance/gait. The original scale included stages 1 through 5. During the 1990s, increments of 0.5 were introduced for some clinical trials. Since then, stage 0 has been added, and stages 1.5 and 2.5 have been proposed. The researchers apply this scale from the observation of the eight-stage clinical picture: 0 (zero) for no signs of the disease; 1 for unilateral disease only; 1.5 for unilateral and axial impairment; 2 for bilateral disease, without impairment of equilibrium; 2.5 for bilateral disease with mild impairment of equilibrium; 3 for bilateral disease with mild and moderate impairment of equilibrium; 4 for severe disability, though capable of standing and walking without help; and 5 for individuals in a wheelchair or confined to bed, in need of full help [35]. The scale is used along with UPDRS for a better assessment of PD. A study showed construct validity was established through moderate to good Spearman rho correlation coefficients with part III of the UPDRS and the H & Y stage (ranging from 0.51 to 0.63) [36].

#### 2.3.3. SEADL

The SEADL scale classifies the level of disability by percent decile ranging from complete independence (100%) to complete dependence (0%); ≤70% indicates the requirement for caregiver assistance. The SEADL reportedly has moderate to very good validity and good reliability [37].

#### 2.3.4. LEDD

The changes in the LEDD at 1 and 5 years post-surgery compared with the preoperative status were also analyzed. The total equivalent dose of levodopa was calculated according to the accepted equivalence between different dopaminergic drugs [38].

#### 2.3.5. PDQ-39

The QoL was assessed using the PDQ-39 (score range, 0–156), which is a disease-specific measurement instrument for patients with PD. A higher score signifies a poorer QoL. It comprises 39 items divided into eight subscales: mobility, activities of daily living (ADL), emotional well-being, stigma, social support, cognition, communication, and bodily discomfort [39]. The PDQ-39 summary index (SI) ranges from 0 to 100. The most optimal estimates for minimal clinically important difference (MCID) thresholds for PDQ-39-SI were −4.72 and +4.22 for detecting minimal clinically important improvement and worsening [40]. Internal consistency reliabilities of the PDQ-39 were greater than 0.70 for all scales, and test-retest reliabilities ranged from 0.68 to 0.94 [41]. The Taiwanese version of the PDQ-39 demonstrated acceptable reliability. The internal consistency reliability was satisfactory for all domains (Cronbach’s alpha ranging from 0.80 to 0.96) except for the social support, cognition, and bodily discomfort domains (Cronbach’s alpha ranging from 0.58 to 0.63). The convergent validity was also supported by strong correlations between domains measuring related constructs of the PDQ-39 and UPDRS (r = 0.81–0.86) [42].

### 2.4. Statistical Analysis

Descriptive analysis was used to assess demographic data, differences, and change rates in UPDRS, SEADL, and PDQ-39 scores, H & Y stage, and LEDD. The difference was defined as post-surgery value − pre-surgery value. The change rate was defined as 100 times (post-surgery value − pre-surgery value)/pre-surgery value. The Kolmogorov–Smirnov test (K-S test) was adopted to evaluate whether the data followed a normal distribution. The comparison of preoperative demographics between male and female patients was analyzed with an independent *t*-test or Wilcoxon rank-sum test, depending on whether it followed a normal distribution. The differences or change rates in longitudinal outcome among three visits were analyzed with a paired *t*-test or Wilcoxon signed-rank test to compare UPDRS, SEADL, and PDQ-39 scores, H & Y stage, and LEDD depending on whether it followed a normal distribution. Missing data were not entered. The data were presented as the means ± standard deviation (SD). A p-value of less than 0.05 was considered a statistical value. All statistical analyses were performed using the SPSS version 25.0 (IBM Corp, Armonk, NY, USA).

## 3. Results

This study included 79 patients with PD who underwent bilateral STN-DBS. All of them achieved more than 30% improvement in motor symptoms in the preoperative levodopa challenge test. Twenty-six patients were lost to follow-up for the following reasons: STN-DBS surgery in other hospitals (*n* = 3), a lack of UPDRS data at 1 year post-surgery (n = 4), a lack of UPDRS data at 5 years post-surgery (*n* = 13), a lack of UPDRS data at 1 and 5 years post-surgery (*n* = 4), and a lack of PDQ-39 data before and after surgery (*n* = 2). On comparing the clinical characteristics of the patients who completed regular visits to date and those lost to follow-up, no significant differences were noted except for gender. Complete data for comparison among preoperational, 1-year, and 5-year follow-ups after STN-DBS surgery were available for 53 patients. The average age when undergoing STN-DBS were 58.89 ± 7.59 with female preponderance (M: F = 24(45%): 29(55%)). The average age of the onset of PD symptoms was 49.53 ± 8.12 years. The mean duration of PD was 9.45 ± 2.91 years. The demographic features and descriptive analyses are shown in Table 1.

### 3.1. Motor Outcome

#### 3.1.1. Off-Medication/On-Stimulation Evaluation

With stimulation in the off-medication state and comparison to the preoperative baseline assessment, there were significant improvements in the UPDRS part II (ADL) scores of 39.71 ± 26.52% and 18.83 ± 37.15% at 1 and 5 years after STN-DBS surgery, respectively. Compared with the baseline, the total UPDRS III (motor) scores at 1 and 5 years were also significantly improved by 36.83 ± 22.51% and 22.75 ± 36.32%, respectively; however, this indicated a slight loss of effectiveness over time (Table 2 and Table 3). In the UPDRS III sub-scale analysis, bradykinesia, tremor, rigidity, and posture-gait symptoms showed significant improvement at 1 and 5 years after surgery. Tremor had the best response in the off-medication/on-stimulation state (69.60 ± 32.48% at 1 year and 67.60 ± 58.23% at 5 years). Axial symptoms improved by 21.64 ± 34.81% at the 1-year follow-up and became even worse than the baseline 5 years after surgery. The H & Y stage significantly improved at 1 year, became worse progressively, and returned to the baseline stage 5 years after surgery. Concerning functional independence for ADL measured with the SEADL scale in the off-medication state at the baseline, only 21 out of 53 (39.62%) patients were completely independent (scoring ≥ 80%). At 1 year post-surgery, there was a statistically significant increase in this percentage (76.71 ± 151.05%), with 45 out of 53 (84.90%) patients being independent, and the percentage (54.82 ± 142.18%) declined but was still significantly higher than baseline, with 31 out of 53 (58.49%) patients scoring ≥ 80% at 5 years post-surgery.

#### 3.1.2. On-Medication/On-Stimulation Evaluation

In comparison with the preoperative on-medication state, there were no significant changes in the UPDRS parts II and III total scores and sub-scales except for a significant improvement in rigidity after 1 year of both on-medication and on-stimulation (Table 2). Moreover, we observed significant deterioration in UPDRS II and UPDRS III total scores, bradykinesia, posture gait, and axial symptoms except for tremor and rigidity (*p* < 0.001) at 5 years after surgery. However, the UPDRS parts II and III total scores after surgery were lower than those in the off-medication/on-stimulation state. A sustained worsening of axial symptoms was observed at 1- and 5-year follow-ups. We found a significant improvement in UPDRS part IV (motor fluctuation and dyskinesia) after surgery (*p* < 0.001). The H & Y stage and SEADL scores in the on-medication state did not significantly differ from the baseline to 1 year post-surgery and worsened after 5 years compared with the baseline (Table 3). Five years after surgery, there was a significant worsening of UPDRS part II (ADL), part III (motor), H & Y stage, and SEADL compared with 1 year in the off-medication and on-medication with stimulation states.

### 3.2. LEDD

In comparison with preoperative LEDD (1139.99 ± 443.34 mg), postoperative LEDD at 1 year (667.58 ± 308.58 mg) and 5 years (842.60 ± 387.10 mg) decreased by 35.32 ± 35.87% (*p* < 0.001) and 15.26 ± 65.76% (*p* < 0.001), respectively (Table 2 and Table 3).

### 3.3. QoL

There was a significant improvement in overall QoL at 1 year post-surgery with a decrease of 14.74 ± 21.37 (22.61 ± 44.30%, *p* < 0.001) in the PDQ-39 total score from the baseline score (57.30 ± 26.29) (Table 4 and Table 5). However, during the long-term follow-up (5 years post-surgery), patients almost completely regressed to baseline levels; the mean PDQ-39 total score was 3.23 points above the preoperative score. Using a cut-off change of −4.72 and +4.22 for detecting MCID in the PDQ-39 SI (see methods), there was a clinically important improvement in QoL in the short term (difference: −9.45 ± 13.70, *p* < 0.001); however, it worsened 5 years after STN-DBS surgery (difference: 2.07 ± 20.92) (Table 4). Dimension scores of mobility, ADL, emotional well-being, stigma, and bodily discomfort were significantly improved from the baseline to 1 year after STN-DBS by 23.33 ± 56.96%, 4.12 ± 191.10%, 8.55 ± 96.84%, 25.50 ± 64.38%, and 28.52 ± 57.13%, respectively. Contrarily, social support, cognition, and communication were not significantly altered at the 1-year follow-up but significantly deteriorated at the 5-year follow-up (Table 5).

## 4. Discussion

Despite optimal pharmacological therapy, approximately 30% of patients with PD experience motor complications after approximately 5 years of treatment [43]. An intolerable variation in QoL is experienced by a substantial proportion of the patients. DBS surgery is the most common symptomatic therapy for PD. The present study aimed to compare the effects of bilateral STN-DBS on motor symptoms and QoL in patients with PD at the baseline and 1 and 5 years post-surgery. Our data showed significant benefits of STN-DBS on motor symptoms and QoL. However, long-term STN-DBS showed variable effects on motor symptoms and QoL in the sample. Our study demonstrated that significant improvements were observed in the UPDRS parts II (ADL) and III (motor) total scores in the off-medication/on-stimulation condition. UPDRS parts II and III were significantly improved by 39.71% and 36.83%, respectively, 1 year after STN-DBS compared with baseline. However, 5 years post-surgery, there was a decline in the therapeutic effects compared with the improvement of 18.83% and 22.75%, respectively, and a slightly diminished effectiveness in the UPDRS part III sub-scales except for tremor. The short-term effects are similar to a recent meta-analysis with 6–12 months follow-up after bilateral STN-DBS implantations; the UPDRS part II score in the off-medication period improved by 47%, and the UPDRS part III score improved by 29.8% [44]. The results are also consistent with previous studies with 5-year follow-up periods [24,25,45,46,47,48]. Tremor is the cardinal sign that benefits the most from STN-DBS. Contrary to medication, the superior efficacy of STN-DBS on tremor amplitude and its beneficial impact on tremor frequency may be explained by the influence of STN-DBS on additional neural circuits independent of dopaminergic neurotransmission [49]. The present results showed that with on-medication/on-stimulation condition, the motor symptoms worsened 5 years post-surgery, as evident from the UPDRS parts II and III total scores, H & Y stage, SEADL score, bradykinesia, posture-gait symptoms, and axial symptoms. A possible cause for the deterioration is decreased levodopa responsiveness [50]. Additionally, with disease progression, levodopa-resistant symptoms, such as axial motor symptoms, might have developed with the longer follow-up period. A previous study demonstrated that treatment with STN-DBS switching from high- to low-frequency stimulation reduced patients’ axial impairments, such as gait, postural stability, and speech, as well as UPDRS motor scores [51]. Since low-frequency stimulation offers less beneficial effects for tremor than high-frequency stimulation, we did not use lower-frequency STN-DBS for improving axial symptoms. Vizcarra et al. demonstrated that treatment with STN-DBS or levodopa alone could reduce the severity of motor symptoms in PD, and co-treatment with STN-DBS and levodopa showed synergistic therapeutic effects [52]. Our results demonstrated a substantial reduction in LEDD and UPDRS part at both 1- and 5-year follow-ups after STN-DBS surgery. The complications related to medication might be reduced, while the levodopa-responsive symptoms of undertreatment could increase [47]. According to DBS follow-up studies from various geographical regions, the preoperative LEDD for Asian patients with advanced PD is 670–1066 mg [47]. The present study obtained higher mean preoperative LEDD (1105.33 ± 413.89 mg), suggesting that the patients might have had more severe disease. LEDD and the severity of PD are recognized risk factors for levodopa-induced dyskinesia and may affect long-term STN-DBS effects on motor functions.

Our results revealed a significant improvement in the overall QoL (22.61 ± 44.30%) and a mean reduction of 14.74 ± 21.37 in the PDQ-39 total score except for the subdomains of social support, cognition, and communication at 1 year; PDQ-39 total scores returned to near baseline levels at 5 years post-surgery. These results are in agreement with those previous studies on QoL after STN-DBS surgery [53,54,55,56]. The reason for the lack of significant improvements in social support may be due to the fact that patients reduce social activities after surgery, especially in our culture, which is more inclined to stay at home during illness. Although STN-DBS reportedly has long-term beneficial effects on QoL [48,57], our findings indicate that the effects of STN-DBS on QoL returned to the baseline 5 years post-surgery. This might be because of levodopa-refractory, stimulation-resistant motor and nonmotor features of PD [23,24]. Additionally, the sustained improvement in QoL could also depend on improvements in other psychosocial aspects of the PDQ-39 scale, including physical, mental, and social domains. QoL might also be affected by factors such as comorbid medical conditions and the patient’s health beliefs and attitudes. All of these factors might influence long-term improvement in QoL. It is important to notice that motor symptoms are not equivalent to QoL, but they do impact QoL following STN-DBS in patients with PD. The study findings will enable clinicians to become more attuned to visible and invisible manifestations of PD that are associated with motor symptoms and QoL through the incorporation of interdisciplinary assessments to better identify the needs of individuals living with this disease.

The risk of developing PD in men is reportedly twice as high as that in women [58]. However, the number of women was proportionately higher in our sample. Female gender proved to be a negative predictor for physical-functioning and socioemotional health-related QoL [58]. The information regarding QoL outcome after STN-DBS can be useful to patients and physicians when counseling patients with PD regarding STN-DBS surgery.

## 5. Limitations

First, we did not have a control group of patients with PD to compare the results of both surgery and non-surgery patients. Our strategy was a pre- and post-surgery comparison in the same patients. Second, 32.91% (26/79) of the patients did not complete the needed assessments or failed to follow through, probably leading to bias and compromising study validity. Patients with unsatisfactory health conditions were more reluctant to participate in follow-up examinations. Third, because of the patients’ unwillingness to turn off the implantable generator of STN-DBS, data were unobtainable for the off-medication and on-medication states with off-stimulation. Fourth, we did not analyze nonmotor symptoms and neuropsychological data in these patients. Fifth, the information regarding the surgery-, device-, and stimulation-related adverse events were not collected. Finally, the QoL measurements in a patient from one time point to the next were not independent, owing to other endogenous and exogenous influences, and \ a dynamic flux between various degrees of health and wellness occurs throughout life. A comprehensive and well-designed study is needed to further investigate the efficacy of STN-DBS in PD.

## 6. Conclusions

The long-term effects of STN-DBS on motor symptoms, especially in tremor and reduction of LEDD, were maintained over 5 years after STN-DBS surgery. The other cardinal motor symptoms, such as axial symptoms, worsened in the long term. The QoL revealed significant improvement in the short term after STN-DBS; however, it tended to regress to preoperative status after 5 years following STN-DBS. However, with the limitation of the study’s design, there was insufficient evidence to determine the long-term effects of STN-DBS on motor symptoms and quality of life in patients with Parkinson’s disease.

## Figures and Tables

**Table 1 healthcare-11-00920-t001:** Preoperative demographics (*n* = 53).

Item	Condition	Female	Male	Total	*p*-Value
Number		29	24	53	
Age of PD onset (yr) †		49.66 (8.16)	49.38 (8.23)	49.53 (8.12)	0.902
Age when underwent STN-DBS (yr) †		59.10 (8.54)	58.63 (6.43)	58.89 (7.59)	0.822
PD duration (yr) †		9.45 (2.91)	9.25 (4.06)	9.36 (3.44)	0.837
LEDD (mg) †		1105.33 (413.89)	1181.88 (482.16)	1139.99 (443.34)	0.537
UPDRS total (0–199) †	off-med/on-stim	78.62 (19.17)	72.25 (19.51)	75.74 (19.4)	0.238
	on-med/on-stim	36.59 (10.68)	37.63 (9.87)	37.06 (10.23)	0.717
Part II (0–52) †	off-med/on-stim	22.21 (7.16)	19.33 (8.01)	20.9 (7.62)	0.174
	on-med/on-stim	7.14 (3.70)	7.83 (3.74)	7.45 (3.70)	0.501
Part III (0–108) †	off-med/on-stim	44.83 (11.64)	42.46 (10.71)	43.75 (11.19)	0.448
	on-med/on-stim	20.31 (7.11)	21.29 (6.25)	20.75 (6.69)	0.600
Part IV (0-23) #		7.00 (2.39)	6.25 (3.38)	6.66 (2.88)	0.349
H & Y stage (0–5) #	off-med/on-stim	3.24 (0.66)	2.92 (0.62)	3.09 (0.66)	0.073
	on-med/on-stim	2.52 (0.37)	2.42 (0.24)	2.47 (0.32)	0.253
SEADL (0–100%) #	off-med/on-stim	57.59 (22.31)	67.5 (23.64)	62.08 (23.23)	0.123
	on-med/on-stim	87.93 (7.74)	90.00 (5.90)	88.87 (6.98)	0.287
PDQ-39 total (0–156) #		61.34 (25.53)	52.42 (26.91)	57.30 (26.29)	0.222
PDQ-39 SI (0–100) #		39.32 (16.37)	33.60 (17.25)	36.73 (16.86)	0.222
Mobility (0–40) †		21.79 (9.32)	16.67 (8.39)	19.47 (9.19)	0.042 *
ADL (0–24) #		8.55 (5.24)	7.42 (5.80)	8.04 (5.48)	0.458
Emotional well-being (0–24) #		9.31 (5.88)	7.71 (5.15)	8.58 (5.57)	0.301
Stigma (0–16) #		5.34 (3.51)	5.13 (4.41)	5.25 (3.90)	0.841
Social support (0–12) #		2.31 (2.69)	2.46 (2.81)	2.38 (2.72)	0.846
Cognition (0–16) #		5.59 (3.21)	5.08 (2.78)	5.36 (3.01)	0.550
Communication (0–12) #		3.03 (2.92)	3.63 (2.37)	3.30 (2.68)	0.430
Bodily discomfort (0–12) †		5.76 (2.60)	4.33 (3.13)	5.11 (2.91)	0.076

PD: Parkinson’s disease; STN-DBS: subthalamic nucleus deep brain stimulation; LEDD: levodopa equivalent daily dose; UPDRS: unified Parkinson’s disease rating scale; H & Y stage: Hoehn & Yahr Stage; SEADL: The Schwab and England Activities of Daily Living; PDQ: Parkinson’s disease questionnaire; SI: summary index; ADL: activities of daily living; off-med: off-medication; on-stim: on-stimulation; on-med: on-medication. Data are presented as mean (standard deviation). †: Independent *t*-test. #: Wilcoxon rank-sum test. * *p*-value < 0.05 was considered statistically significant after test.

**Table 2 healthcare-11-00920-t002:** UPDRS, H & Y stage, SEADL, LEDD preoperative assessment and at 1 year, 5 years follow-up after surgery (*n* = 53).

Item	Condition	Pre-OP	Post-OP 1Y	Diff. (1Y vs. Pre-OP)	*p*-Value	Post-OP 5Y	Diff. (5Y vs. Pre-OP)	*p*-Value
UPDRS total †	off-med/on-stim	75.74 (19.4)	45.60 (14.63)	−30.13 (18.55)	<0.001 *	56.06 (16.98)	−19.68 (22.19)	<0.001 *
	on-med/on-stim	37.06 (10.23)	34.91 (10.60)	−2.15 (10.95)	0.159	44.42 (14.16)	7.36 (14.67)	0.001 *
Part II †	off-med/on-stim	20.91 (7.62)	11.94 (5.41)	−8.96 (6.57)	<0.001 *	15.75 (6.86)	−5.15 (7.94)	<0.001 *
	on-med/on-stim	7.45 (3.70)	8.08 (3.68)	0.62 (3.69)	0.224	11.79 (6.30)	4.34 (5.80)	<0.001 *
Part III †	off-med/on-stim	43.75 (11.19)	26.23 (8.07)	−17.53 (11.11)	<0.001 *	31.32 (9.54)	−12.43 (13.05)	<0.001 *
	on-med/on-stim	20.75 (6.69)	20.11 (6.94)	−0.64 (6.80)	0.495	25.15 (7.79)	4.40 (8.53)	<0.001 *
Bradykinesia †	off-med/on-stim	18.43 (4.63)	12.92 (3.54)	−5.51 (5.36)	<0.001*	14.74 (4.13)	−3.70 (5.80)	<0.001 *
	on-med/on-stim	10.23 (3.46)	11.00 (3.68)	0.77 (3.78)	0.142	12.43 (3.52)	2.21 (4.32)	<0.001 *
Tremor #	off-med/on-stim	7.19 (5.91)	2.68 (3.29)	−4.51 (4.53)	<0.001 *	1.72 (2.33)	−5.47 (5.22)	<0.001 *
	on-med/on-stim	1.34 (2.25)	1.11 (1.67)	−0.23 (2.32)	0.480	0.51 (1.20)	−0.83 (2.49)	0.019 *
Rigidity #	off-med/on-stim	8.09 (3.32)	3.51 (2.41)	−4.58 (3.65)	<0.001 *	5.75 (2.93)	−2.34 (4.26)	<0.001 *
	on-med/on-stim	3.81 (2.84)	2.40 (2.14)	−1.42 (2.85)	0.001 *	4.42 (2.63)	0.60 (3.39)	0.201
Posture-gait #	off-med/on-stim	3.77 (1.45)	2.60 (1.10)	−1.17 (1.31)	<0.001 *	3.32 (1.24)	−0.45 (1.53)	0.035 *
	on-med/on-stim	1.92 (1.14)	2.09 (1.11)	0.17 (1.09)	0.261	2.77 (1.15)	0.85 (1.23)	<0.001*
Axial symptoms †	off-med/on-stim	8.34 (2.92)	6.06 (2.49)	−2.28 (2.85)	<0.001 *	7.87 (2.85)	−0.47 (3.25)	0.296
	on-med/on-stim	4.40 (2.00)	4.81 (2.18)	0.42 (1.83)	0.105	6.70 (2.55)	2.30 (2.21)	<0.001 *
Part IV #		6.66 (2.88)	4.28 (2.35)	−2.38 (3.54)	<0.001 *	4.98 (2.43)	−1.68 (3.01)	<0.001 *
H & Y stage #	off-med/on-stim	3.09 (0.66)	2.67 (0.42)	−0.42 (0.71)	<0.001 *	3.04 (0.59)	−0.06 (0.89)	0.646
	on-med/on-stim	2.47 (0.32)	2.50 (0.39)	0.03 (0.37)	0.582	2.85 (0.48)	0.38 (0.45)	<0.001 *
SEADL (%) #	off-med/on-stim	62.08 (23.23)	83.02 (11.19)	20.94 (20.69)	<0.001*	72.08 (19.05)	10.00 (26.38)	0.008 *
	on-med/on-stim	88.87 (6.98)	88.30 (7.53)	−0.57(7.18)	0.569	81.51 (14.46)	−7.36 (14.96)	0.001 *
LEDD (mg) †		1139.99 (443.34)	667.58 (308.58)	−472.41(391.72)	<0.001 *	842.60 (387.10)	−297.39 (466.69)	<0.001 *

UPDRS: unified Parkinson’s disease rating scale; H & Y stage: Hoehn & Yahr Stage; SEADL: The Schwab and England Activities of Daily Living; LEDD: levodopa equivalent daily dose; Diff: difference; off-med: off-medication; on-stim: on-stimulation; on-med: on-medication; OP: operation; Diff: difference. Data are presented as mean (standard deviation). †: Paired *t*-test. #: Wilcoxon signed-rank test. * *p* < 0.05 was considered statistically significant after test.

**Table 3 healthcare-11-00920-t003:** Change rates of UPDRS II, III, H & Y stage, SEADL, and LEDD after surgery (*n* = 53).

Item	Condition	N	Change Rate (%) (1Y vs. Pre-OP)	*p*-Value	Change Rate (%) (5Y vs. Pre-OP)	*p*-Value
UPDRS Part II †	off-med/on-stim	53	−39.71 (26.52)	<0.001 *	−18.83 (37.15)	<0.001 *
	on-med/on-stim	51	33.08 (122.37)	0.224	90.97 (156.78)	<0.001 *
UPDRS Part III †	off-med/on-stim	53	−36.83 (22.51)	<0.001 *	−22.75 (36.32)	<0.001 *
	on-med/on-stim	53	3.58 (47.37)	0.495	36.54 (103.27)	<0.001 *
Bradykinesia †	off-med/on-stim	53	−24.78 (30.79)	<0.001 *	−13.24 (43.67)	<0.001 *
	on-med/on-stim	53	29.5 (141.05)	0.142	66.29 (301.48)	<0.001 *
Tremor #	off-med/on-stim	47	−69.6 (32.48)	<0.001 *	−67.61 (58.23)	<0.001 *
	on-med/on-stim	27	18.56 (136.29)	0.480	−68.83 (82.03)	0.019 *
Rigidity #	off-med/on-stim	52	−50.69 (35.59)	<0.001 *	−18.9 (55.76)	<0.001 *
	on-med/on-stim	49	−14.25 (120.91)	0.001*	42.9 (132.23)	0.201
Posture-gait #	off-med/on-stim	53	−24.14 (33.48)	<0.001*	−1.62 (47.96)	0.035 *
	on-med/on-stim	48	20.87 (76.12)	0.261	61.67 (83.87)	<0.001 *
Axial symptoms †	off-med/on-stim	53	−21.64 (34.81)	<0.001 *	2.63 (44.4)	0.296
	on-med/on-stim	52	17.28 (65.34)	0.105	77.23 (100.46)	<0.001 *
H & Y stage #	off-med/on-stim	53	−10.53 (20.93)	<0.001 *	2.23 (28.66)	0.646
	on-med/on-stim	53	1.89 (15.99)	0.582	16.07 (18.77)	<0.001 *
SEADL #	off-med/on-stim	53	76.71 (151.05)	<0.001 *	54.82 (142.18)	0.008 *
	on-med/on-stim	53	−0.36 (8.34)	0.569	−7.91 (17.25)	0.001 *
LEDD †		53	−35.52 (35.87)	<0.001 *	−15.26 (65.76)	<0.001 *

UPDRS: unified Parkinson’s disease rating scale; H & Y stage: Hoehn & Yahr Stage; SEADL: The Schwab and England Activities of Daily Living; LEDD: levodopa equivalent daily dose; off-med: off-medication; on-stim: on-stimulation; on-med: on-medication; OP: operation. Data are presented as mean (standard deviation). †: Paired *t*-test. #: Wilcoxon signed-rank test. * *p* < 0.05 was considered statistically significant after test.

**Table 4 healthcare-11-00920-t004:** PDQ-39 preoperative assessment and at 1- and 5-year follow-ups after surgery (*n* = 53).

Item	Pre-OP	Post-OP 1Y	Diff. (1Y vs. Pre-OP)	*p*-Value	Post-OP 5Y	Diff. (5Y vs. Pre-OP)	*p*-Value
PDQ-39 total #	57.3 (26.29)	42.57 (25.76)	−14.74 (21.37)	<0.001 *	60.53 (33.06)	3.23 (32.63)	0.475
PDQ-39 SI #	36.73 (16.86)	27.29 (16.51)	−9.45 (13.7)	<0.001 *	38.8 (21.19)	2.07 (20.92)	0.475
Mobility †	19.47 (9.19)	13.58 (9.94)	−5.89 (9.73)	<0.001 *	19.96 (11.15)	0.49 (11.36)	0.754
ADL #	8.04 (5.48)	4.81 (4.93)	−3.23 (5.54)	<0.001 *	8.47 (6.34)	0.43 (7.30)	0.667
Emotional well-being #	8.58 (5.57)	6.74 (5.20)	−1.85 (4.26)	0.003 *	8.70 (5.98)	0.11 (5.90)	0.889
Stigma #	5.25 (3.90)	3.77 (3.68)	−1.47 (3.20)	0.002 *	4.66 (4.75)	−0.58 (4.37)	0.334
Social support #	2.38 (2.72)	2.57 (2.76)	0.19 (2.86)	0.633	3.81 (3.23)	1.43 (3.38)	0.003 *
Cognition #	5.36 (3.01)	5.06 (3.38)	−0.30 (2.75)	0.428	6.68 (3.59)	1.32 (4.11)	0.023 *
Communication #	3.30 (2.68)	3.28 (2.78)	−0.02 (2.58)	0.958	4.83 (3.27)	1.53 (3.42)	0.002 *
Bodily discomfort †	5.11 (2.91)	3.32 (2.46)	−1.79 (3.18)	<0.001 *	4.17 (3.11)	−0.94 (3.31)	0.043 *

PDQ: Parkinson’s disease questionnaire; SI: summary index; ADL: activities of daily living; OP: operation; 1Y: 1 year; Diff: difference; 5Y: 5 years. Data are presented as mean (standard deviation). †: Paired *t*-test. #: Wilcoxon signed-rank test. * *p* < 0.05 was considered statistically significant after test.

**Table 5 healthcare-11-00920-t005:** Change rates of PDQ-39 after surgery (*n* = 53).

Item	N	Change Rate (%) (1Y vs. Pre-OP)	*p*-Value	Change Rate (%) (5Y vs. Pre-OP)	*p*-Value
PDQ-39 total #	53	−22.61 (44.30)	<0.001 *	15.52 (66.60)	0.475
PDQ-39 SI #	53	−22.60 (44.30)	<0.001 *	15.52 (66.61)	0.475
Mobility †	53	−23.33 (56.96)	<0.001 *	20.26 (81.16)	0.754
ADL #	51	−4.12 (191.10)	<0.001 *	60.94 (210.16)	0.667
Emotional well-being #	52	−8.55 (96.84)	0.003 *	14.06 (77.94)	0.889
Stigma #	47	−25.50 (64.38)	0.002 *	−3.48 (117.29)	0.334
Social support #	34	4.31 (129.60)	0.633	69.76 (198.42)	0.003 *
Cognition #	53	11.43 (135.80)	0.428	67.08 (167.92)	0.023 *
Communication #	45	−1.54 (75.97)	0.958	65.18 (108.58)	0.002 *
Bodily Discomfort †	49	−28.57 (57.13)	<0.001 *	−10.56 (62.90)	0.043 *

PDQ: Parkinson’s disease questionnaire; OP: operation; SI: summary index; ADL: activities of daily living; 1Y: 1 year; OP: operation; 5Y: 5 years. Data are presented as mean (standard deviation). †: Paired *t*-test. #: Wilcoxon signed-rank test. * *p* < 0.05 was considered statistically significant after test.

## Data Availability

The data presented in this study is not publicly available because of privacy restrictions.

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
