# Peer review of "Long-Term Effects of Subthalamic Stimulation on Motor Symptoms and Quality of Life in Patients with Parkinson’s Disease"

_healthcare, 2023, doi:10.3390/healthcare11060920_

Round 1

Reviewer 1 Report

This manuscript shows the long-term effects of deep brain stimulation on motor function and quality of life in patients with Parkinson's disease, which is an interesting topic. But, as the authors note, the study lacked a comparison with the non-surgical group, nor did it look at non-motor disorders, which can also have a significant impact on patients' quality of life. Despite the limitations of this study, we believe that the data in this study are consistent with other reports, indicating that deep brain stimulation plays an important role in the treatment of Parkinson's disease and is still valuable for reference. Therefore, we believe that this manuscript can be accepted, but the following problems need to be solved first:

- Add patient age, gender and other information

- We expect the authors to go into detail about all of the methods covered in Part 2.3

- Despite the simplicity of the study, we hope the authors can add relevant schematic

- What are the advantages and disadvantages of deep brain stimulation compared to other treatment strategies? We hope the authors supplement related review

- Figure 1 Need to supplement statistical analysis

Author Response

Reviewer 1

Comments and Suggestions for Authors

This manuscript shows the long-term effects of deep brain stimulation on motor function and quality of life in patients with Parkinson's disease, which is an interesting topic. But, as the authors note, the study lacked a comparison with the non-surgical group, nor did it look at non-motor disorders, which can also have a significant impact on patients' quality of life. Despite the limitations of this study, we believe that the data in this study are consistent with other reports, indicating that deep brain stimulation plays an important role in the treatment of Parkinson's disease and is still valuable for reference. Therefore, we believe that this manuscript can be accepted, but the following problems need to be solved first:

- Add patient age, gender and other information

Response: Thanks for your suggestion. We have added age when underwent STN-DBS and sex in Table 1 (line 205) and in Results section paragraph 1(lines 199-201).

- We expect the authors to go into detail about all of the methods covered in Part 2.3

Response: The authors have added supplementary explanations in this section. Please see Materials and Methods in Part 2.3. (lines117-174)

- Despite the simplicity of the study, we hope the authors can add relevant schematic

Response: The following schematic is the research steps of this study.(Please see the attached file)

- What are the advantages and disadvantages of deep brain stimulation compared to other treatment strategies? We hope the authors supplement related review

Response: The authors have added the advantages and disadvantages of deep brain stimulation compared to medical therapy. In introduction, paragraph 1: “DBS offers several advantages over other surgical approaches for neuromodulation. These advantages include the capacity to titrate stimulation parameters to maximize benefits and reduce adverse effects and the opportunity to directly interface with the circuit pathology that drives overt symptoms (Lozano, A.M.---et al., 2019). A systematic review showed DBS was superior to the best medical therapy at improving impairment/disability, QoL, and reducing medication doses, but these benefits need to be weighed against the higher risk of serious adverse events of surgery (Bratsos, S.---,2018).”  (lines 40-46)

- Figure 1 Need to supplement statistical analysis

Response: The authors have added the statistical analysis used in Figure 1. Please refer to the figure legend of Figure 1. (The lines from bottom to top represented minimum, first quartile, median, third quartile, and maximum accordingly. Paired t-test was adopted to determine whether significant mean difference existed after surgery.)

Reviewer 2 Report

The aim of the paper is to compare motor symptoms and quality of life in a large group of Asian patients with PD before and following up at one and five years post STN-BDS surgery.

For Figure 1, the lines should be explained. It would be helpful to include all of the data points to give a clearer communication of the results, and even include connecting lines to illustrate the range of benefits seen.

Author Response

Long-term effects of subthalamic stimulation on motor symptoms and quality of life in patients with Parkinson’s disease

Reviewer 2

Comments and Suggestions for Authors

The aim of the paper is to compare motor symptoms and quality of life in a large group of Asian patients with PD before and following up at one and five years post STN-BDS surgery.

For Figure 1, the lines should be explained. It would be helpful to include all of the data points to give a clearer communication of the results, and even include connecting lines to illustrate the range of benefits seen.

Response: The authors have specified the meaning of the lines in Figure 1. Please refer to the figure legend of Figure 1. (The lines from bottom to top represented minimum, first quartile, median, third quartile, and maximum accordingly. Paired t test was adopted to determine whether significant mean difference existed after surgery.) However, our statistical software did not allow us to display all data points or add additional lines to the figure.

Reviewer 3 Report

Long-term effects of subthalamic stimulation on motor symptoms and quality of life in patients with Parkinson’s disease

General comments

I believe this is an important topic and more research is needed in this area. I value the effort of the authors and the work done since carrying out a study with a population with neurodegenerative diseases is always complicated but at the same time it is very relevant and important to be able to improve the quality of life of these patients.

I will now offer some comments on their work.

One of the most important limitations of their study is the lack of a control group. Having a control group would have been essential to be able to interpret their results with greater certainty and rigor.

Specific comments

Abstract

The conclusion cannot be drawn directly from the results. The conclusion must be a conclusion of its own in relation to the study.

Introduction

In line 55: Self-assessed health status may provide a more comparable predictor of outcome 55 than many objective measures of health.

Do the authors have a reference for this statement, or is it a hypothesis of the authors?

Methodology

Line 76: you say in this line, that changes in motor symptoms and quality of life were evaluated in PD patients from baseline to 5 years after surgery. However, in other sections of the article you comment that changes were evaluated at baseline, 1 and 5 years. Clarify this point.

Line 91: reference the MRI machine. Has the imaging protocol been used previously in other studies? Why was this protocol used?

Line 96: reference the browser

Line 101: reference Leksell G-frame.

Line 107: "The final trajectory for electrode implantation (Medtronic 107 3389 DBS leads) was selected based on adequate length of STN hyperactivity neuronal 108 firing and the presence of movement-related changes in firing pattern." why did the authors rely on this method to choose the electrode implantation trajectory? Is it the most commonly used? Do other studies use it?

It would be advisable to inform the reader if the assessment instruments, scales, etc. are valid scales in this population, as well as their validity or reliability has been reported. Thanks for the effort to the authors because in some of them they do comment and reference it.

Statistical analysis

Was the normality of the data evaluated?

It would be interesting to provide some measure of the size of the effect

Results:

Why do they compare the change rate and not the mean scores? Furthermore, it is not described in the statistical analysis section how this change rate is going to be calculated. Why is it expressed in percentages?

I do not understand very well what this variable expresses and how it is calculated.

It would be advisable for the tables to be in Word format, not in image format.

Regarding the presentation of the data, I consider that it is not necessary to put the two symbols, () and ±, i.e., the authors can put 44.2 ± 1.2 or 44.2 (1.2).

In table 1, the authors mention in the table footnote that the data are presented as percentages but none of the variables I believe are in percentages.

In Table 2, there is a blank jump between SEADL and LEDD.

Discussion

LINE 344: The risk of developing PD in men is reportedly twice as high as in women [ [50]].

I think there is one too many []].

The last part of the conclusion, from line 329, I think is not a conclusion that can be drawn from your study.

Author Response

Long-term effects of subthalamic stimulation on motor symptoms and quality of life in patients with Parkinson’s disease

Reviewer 3

General comments

I believe this is an important topic and more research is needed in this area. I value the effort of the authors and the work done since carrying out a study with a population with neurodegenerative diseases is always complicated but at the same time it is very relevant and important to be able to improve the quality of life of these patients.

I will now offer some comments on their work.

One of the most important limitations of their study is the lack of a control group. Having a control group would have been essential to be able to interpret their results with greater certainty and rigor.

Response: Thanks for your reminder. Because of cultural and ethical issues, we did not adopt the experimental and control groups' research design. We did not have a control group of patients with PD to compare with the results of both surgery and non-surgery patients. These findings of this study need to treat circumspectly.

Specific comments

1.Abstract

The conclusion cannot be drawn directly from the results. The conclusion must be a conclusion of its own in relation to the study.

Response: The authors have added the conclusion in the abstract and lines 25-27: ”The long-term effects of STN-DBS on motor symptoms were maintained over 5 years after STN-DBS surgery. At the same time, STN-DBS had no long-lasting effect on QoL.”

2.Introduction

In line 55: Self-assessed health status may provide a more comparable predictor of outcome than many objective measures of health. Do the authors have a reference for this statement, or is it a hypothesis of the authors?

Response: It is a hypothesis of the authors.

3.Methodology

(1) Line 76: you say in this line, that changes in motor symptoms and quality of life were evaluated in PD patients from baseline to 5 years after surgery. However, in other sections of the article you comment that changes were evaluated at baseline, 1 and 5 years. Clarify this point.

Response: The authors have revised the descriptions. This can be seen in Materials and Methods paragraph 1: “The current study utilized a prospective, observational cohort design to evaluate changes in motor symptoms and QoL in patients with PD before surgery and at 1 and 5 years after STN-DBS surgery.” (lines 84-85)

(2)Line 91: reference the MRI machine. Has the imaging protocol been used previously in other studies? Why was this protocol used?

(3)Line 96: reference the browser

(4)Line 101: reference Leksell G-frame.

(5)Line 107: "The final trajectory for electrode implantation (Medtronic 107 3389 DBS leads) was selected based on adequate length of STN hyperactivity neuronal 108 firing and the presence of movement-related changes in firing pattern." why did the authors rely on this method to choose the electrode implantation trajectory? Is it the most commonly used? Do other studies use it?

Response: The authors have revised the descriptions of 2.2. Surgical Procedure. The details of surgical procedures have been published in our several studies. We thank you for your reminder and for adding these references. (lines 100-114)

(6) It would be advisable to inform the reader if the assessment instruments, scales, etc. are valid scales in this population, as well as their validity or reliability has been reported. Thanks for the effort to the authors because in some of them they do comment and reference it.

Response: The authors have revised the descriptions of outcome measurement. In measure 2.3.1. UPDRS, 2.3.2. H & Y Stage and 2.3.5. PDQ-39 of the revised manuscript.

4.Statistical analysis

(1) Was the normality of the data evaluated?

(2) It would be interesting to provide some measure of the size of the effect

Response: The authors have specified that we adopted Kolmogorov-Smirnov test (K-S test) to evaluate whether the data followed normal distribution in Method section. Please refer to 2.4. Statistical Analysis paragraph 1. (lines 180-181)

5.Results:

(1) Why do they compare the change rate and not the mean scores? Furthermore, it is not described in the statistical analysis section how this change rate is going to be calculated. Why is it expressed in percentages?

(2) I do not understand very well what this variable expresses and how it is calculated.

Response: The authors have specified the definition of difference and change rate in 2.4. Statistical Analysis paragraph 1. (lines 178-180)

The change rate was defined as 100*(post-surgery value – pre-surgery value)/pre-surgery value. We calculated change rates to compare our results with former studies.

(3) It would be advisable for the tables to be in Word format, not in image format.

Response: The authors will submit the tables in Word format to the assistant editor for help.

(4) Regarding the presentation of the data, I consider that it is not necessary to put the two symbols, () and ±, i.e., the authors can put 44.2 ± 1.2 or 44.2 (1.2).

Response: The authors have changed the presentation method of the data in Table 1~5.

(5) In table 1, the authors mention in the table footnote that the data are presented as percentages but none of the variables I believe are in percentages.

Response: Thanks for your reminder. The authors have corrected it. Please refer to the footnote of Table 1 of the revised manuscript.

(6) In Table 2, there is a blank jump between SEADL and LEDD.

Response: The authors have corrected it.

6.Discussion

(1) LINE 344: The risk of developing PD in men is reportedly twice as high as in women [ [50]]. I think there is one too many []].

Response: The authors have corrected it.

(2) The last part of the conclusion, from line 329, I think is not a conclusion that can be drawn from your study.

Response: The authors have removed the last part of the conclusion: “The study findings will enable clinicians to become more attuned to visible and invisible manifestations of PD that are associated with motor symptoms and QoL through the incorporation of interdisciplinary assessments to better identify the needs of individuals living with this disease.”

Round 2

Reviewer 3 Report

The authors have made a great effort and have provided clarification on all points.

I would like to congratulate the authors as their work has been substantially improved.